# Utilization of diabetes self-management program among breast, prostate, and colorectal cancer survivors: Using 2006–2019 Texas Medicare data

Wei-Chen Lee[1]*, Biai Dominique Elmir Digbeu[2], Hani Serag[3], Hanaa Sallam[3], Yong-Fang Kuo[2]

1 Department of Family Medicine, University of Texas Medical Branch, Galveston, Texas, United States of America, 2 Department of Biostatistics & Data Science, University of Texas Medical Branch, Galveston, Texas, United States of America, 3 Department of Internal Medicine, University of Texas Medical Branch, Galveston, Texas, United States of America

* weilee@utmb.edu

**Data Availability Statement:** The research was focused on secondary data analysis of 100% Texas Medicare claims with information on patient's demographics, diagnosis, prescription, procedure,

## Abstract

### Background

Cancer treatment is associated with inferior health outcomes such as diabetes. Medicare provides Diabetes Self-Management Training (DSMT) program to beneficiaries to achieve normal metabolic control and reduce the risk of micro and macro-vascular complications. This study aimed to examine the trend of DSMT utilization among cancer survivors and assess individual characteristics associated with it.

### Methods

The data for this study was from Texas Cancer Registry-Medicare linkage data of patients with prostate, breast, or colorectal cancer diagnosed in 1999–2017. Outcome variables include the number of first-time DSMT users, the number of total users, and the average number of DSMT utilization in minutes. We performed logistic regression and gamma regression to obtain a multivariable-adjusted odds ratio for the association between DSMT utilization and individual characteristics.

### Results

The number of first-time users has slowly increased over the years (from 99 to 769 per 1,000) but suddenly dropped after 2016. The number of all users (first-time and follow-up users) has increased (from 123 to 1,201 per 1,000) and plateaued after 2016. Determinants including Hispanic ethnicity (O.R. = 1.10) and Medicare-Medicaid dual eligibility (O.R. = 1.25) are positively associated with both the initiation and retention of the DSMT. A barrier to both initiation and retention of DSMT is living in a metropolitan area (O.R. = 0.90).

physician's characteristics, and facility type. The existing data is available from Centers to Medicare and Medicaid Services (CMS) under the specific data user agreement (DUA). Data cannot be shared publicly due to the risk of potential identification of individual patients. To acquire the same type of data, investigators need to submit an application to the CMS. All researchers interested in CMS data can find introduction of each data file, data dictionary, and user agreement on the Research Data Assistance Center (ResDAC). It also provides assistance to academic, non-profit, for-profit, and government researchers. More information can be found on the CMS website (https://www.cms.gov/research-statistics-data-and-systems/files-for-order/data-disclosures-data-agreements/overview).

**Funding:** The study was made possible by the grant (#RP210130) from Cancer Prevention and Research Institute of Texas (CPRIT) Data Management and Analysis Core (DMAC). The funder had no role in study design, data collection and analysis, decision to public, or preparation of the manuscript.

**Competing interests:** The authors have declared that no competing interests exist.

## Conclusions

Multi-level strategies to enhance accessibility and availability of DSMT programs for Medicare beneficiaries are highly recommended. Examining the determinants of initiation and retention of DSMT over 14 years provides insights on strategies to meet the needs of cancer survivors and reduce the burden of diabetes on them.

## Introduction

Diabetes is a growing public health concern affecting 29.1 million people in the United States [1]. It continues to be one of the top ten causes of death, and accounts for $245 billion in financial loss. From 1999 to 2020, the percentage of adults in Texas who reported having diabetes has been higher than the nation's average (Texas:12.0% vs. the U.S.: 10.0%) [2]. Aging is an independent risk factor for developing diabetes [3]. Only 13.8% of adults aged 45–64 have diabetes; however, more than 21.4% of adults aged 65 and above have diabetes [4]. In addition, 27.5% of Medicare fee-for-service (FFS) beneficiaries have been diagnosed with diabetes in 2019 according to the Centers for Medicare and Medicaid Services (CMS) Chronic Data Warehouse [5]. In light of this public health concern, it is imperative to implement evidence-based interventions and prevent or delay the onset of further complications and comorbidities, including kidney disease and cardiovascular diseases [1, 4].

Diabetes Self-Management Training (DSMT), also known as Diabetes Self-Management Education and Support (DSMES), is a Medicare preventive service that helps beneficiaries to manage their diabetes, prevent additional complications, and reduce hospitalizations and medical expenditure [6, 7]. CMS approves two national DSMT to accredit DSMT entities including The American Diabetes Association (ADA) and The Association of Diabetes Care & Education Specialists (ADCES) [6]. Both accreditation organizations can collect and record patient's information quarterly such as diabetic condition duration and last eye exam date to assess patient's outcomes after receiving DSMT/DSMES. DSMES contains medically necessary education supplied by certified providers to beneficiaries with Type 1 or Type 2 diabetes and reimbursed by Medicare Part B [8]. Although Section 4105(a) of the Balanced Budget Act of 1997 has permitted Medicare coverage of DSMES since 1997, DSMES services are not widely known or used. A 2014 study of private insurance data shows that 6.8% of clients participated in DSMES, slightly higher than a 2015 study of Medicare data which is only 5% [7, 9]. DSMES referrals were reported to be higher among Medicare beneficiaries than those covered by private insurance, yet this did not necessarily lead to DSMT/DSMES utilization [10].

With the advancement of cancer therapies over the last decade, the population of cancer survivors in the US has expanded to over 18 million and is projected to increase to 22.5 million in the next ten years [11]. Among people aged 65 and older, all cancer incidences decreased by 1.2 per 100,000 people overall in the U.S. from 2014 to 2018 but only decreased by 0.6 in Texas [12]. The diabetes risk is increased among cancer survivors that the development of diabetes in cancer survivors was 1.39 times higher than noncancer individuals [13, 14]. A great number of studies emphasized the benefits of healthy diet and physical activity to stop the recurrence of cancer in the survivors such as the overall survival was 55% in the group with good control and only 23% in the poor control group; however, none of them investigated the initiation and retention of using DSMES programs [15–18]. A study further points out the important role of oncology providers and counselors to advocate for nonpharmacological approaches and healthy lifestyle changes to help survivors cope with the burden of cancer [16]. To increase the

utilization of innovative DSMES that strengthens the wellness of cancer survivors in Texas, this study retrospectively examined the number of cancer survivors with diabetes who were first-time users of DSMES, the accumulated number of cancer survivors who had ever used DSMES, and the average length of time with DSMES. In addition, the study investigated the contributing factors of DSMES utilization including demographics, socioeconomic status, and health conditions. The understanding of utilization level among cancer survivors may enable policy makers to formulate public health recommendations. The finding of this study may also inform a more coordinated care model among oncology, endocrinology, and primary care and identify the importance of the reimbursement guideline to incentivize DSMES utilization and improve cancer survivorship.

## Materials and methods

### Data source

This is a retrospective cohort study using Texas Cancer Registry (TCR)-Medicare linkage data with cancer patients diagnosed from 1999–2017. TCR is one of the largest cancer registries in the U.S; data collected by TCR include patient demographics, primary tumor site, stage, first course of treatment, tumor morphology, cause of death, and survival. TCR-Medicare linkage was performed by the National Cancer Institute (NCI) and the CMS. The Medicare claims data include billing information on hospital stays, physician services, and hospital outpatient visits. For this study, data were extracted from Medicare beneficiary summary files, Medicare Provider Analysis and Review files, Outpatient Standard Analytical Files, and Medicare Carrier files. The study involving human subjects was reviewed and approved by IRB (#21–0311) of the University of Texas Medical Branch at Galveston and another IRB (#22–0031) for Medicare data overall. This data is acquired through the Data Use Agreement (DUA) with the NCI and individual investigators are not required to obtain participant consent. Medicare Beneficiary Identifiers (MBI) were used to uniquely identify Medicare patients and link their clinical claims across multiple years.

### Study cohort

This study identified cancer patients first time diagnosed with breast, prostate, or colorectal cancer at any time during the period of 1999–2014 and who had survived over five years since diagnosis following the methodology used in our previous study [19]. A yearly cohort from 2006 to 2019 was generated (14 cohorts in total). To account for at least 5-year survival patients who were diagnoses with cancer between 1999 and 2001 were included in the 2006 cohort; patients who were diagnosed with cancer between 1999 and 2002 were included in the 2007 cohort, and vice versa until they dropped out of Medicare. The sample collection was repeated every year and the 2019 cohort included patients who were diagnosed with cancer between 1999 and 2014.

At each study year, individuals who were under 66 years old, had passed away, were living in a nursing home, or had not enrolled in both Medicare Parts A & B for 12 consecutive months in the year were excluded. The study cohort was only limited to cancer survivors with at least two outpatient claims, or one inpatient claim related to diabetes (ICD-9 and ICD-10: 250.x, E10.x, E11.x) in the period of 1999 to 2014. For instance, the 2006 cohort included patients with diabetes claims between 1999 and 2006 and the 2007 cohort included patients who had diabetes-related claims between 1999 and 2007. The same process was repeated to select cohorts that met the inclusion criteria. However, beneficiaries with multiple cancers, who lived in long-term care facility, who were diagnosed with end-stage renal disease or

kidney transplantation, or who had enrolled in Medicare Advantage were excluded. S1 Fig in S1 File provides a flow chart of eligible cohort in 2006 as an example.

## Dependent variables

The primary outcome was the first-time use of DSMES per 1000 in each year from 2006 to 2019 (see Formula 1). In the multivariable model, the outcome was a dichotomous variable including 1 = yes (first time) and 0 = no (not the first-time use). The secondary outcome was the number of minutes of DSMES that a patient has received (see Formula 2). The service can be covered by Medicare Part B using either Healthcare Common Procedure Code System [HCPCS] code G0108 or G0109 [20]. The HCPCS code G0108 refers to diabetes outpatient self-management training services for an individual per 30 minutes and G0109 refers to DSMES for a group (two or more people) per 30 minutes. Within the first 12 months, Medicare beneficiaries can attend no more than 10 hours of training in a group setting focused on increasing patients' self-care knowledge and skills. CMS reimburses no more than two hours of follow-up training for subsequent years. Based on the formula below, 28 rates were generated and used to constitute two trends from 2006 to 2019.

2006 cohort: First-time DSMES per 1000 = (# of first-time DSMES users in 2006) / (# of all cancer survivors in 2006) *1,000 (1)

2006 cohort: Mean DSMES utilization time in minutes = [(sum of # of minutes of first-time users) + (sum of # of minutes of follow-up users in 2006)] / (# of first-time users + # of follow-up users in 2006) (2)

## Independent variable & covariates

The primary independent variable of this study is the year of attending DSMES. The covariates were categorized into two parts: demographic characteristics and health conditions. Demographic factors included each beneficiary's age, sex (male/female), race/ethnicity (White, Black, Hispanic, and Other), Medicare-Medicaid dual eligibility (yes/no), education, poverty, and income at the zip code level, and residence (metro/ non-metro area). The encrypted zip code data was created by TCR Medicare using the Census 1990, Census 2000, and the American Community Survey. Metro/non-metro was classified based on the Rural-Urban Continuum codes for years 1993, 2003, and 2013, and more information can be found on the website [21]. Health conditions included medical comorbidities and type of diabetes. The study adapted the Elixhauser Comorbidity Index to exclude diabetes and assess the burden of comorbidities [22]. Next, functional status was defined using each patient's Medicare original entitlement to disability insurance (disabled or not disabled). Finally, patient's diagnoses were classified as Type 1 diabetes (ICD-9 250.x1, ICD-9 250.x3, and ICD-10 E10.xx), Type 2 diabetes (ICD-9 250.x0, ICD-9 250.x2, and ICD-10 E11.xx), and diabetes with complications [23, 24].

## Statistical analyses

Trends (2006–2019) of DSMES utilization (first-time users and follow-up users) among yearly study cohorts including patients who survived cancer for at least five years and living with Type 1 or Type 2 diabetes were generated. For each year from 2006 to 2019, patients' demographics, type of cancer survived, presence of incident Type 1 or Type 2 diabetes, presence of uncontrolled Type 1 or Type 2 diabetes, DSMES utilization (first-time users, follow-up users, time of utilization), Medicare original entitlement, dual eligibility, education, poverty, residence and Elixhauser comorbidity were summarized using frequency and percentage for categorical variables and mean and standard deviation for continuous variables. Multivariable logistic regression to predict the likelihood of first-ever DSMES utilization (yes = the first

time/no = not the first time) was performed using the same set of the covariates but excluding the poverty level, which is highly correlated with the education level. The study has repeated the model predicting the first-time utilization by three types of cancers (S1-S3 Tables in S1 File). Because every cancer survivor could start DSMES at any given time from 1999 to 2019, the accumulated number of minutes of DSMES among all users was used as the secondary outcome variable. A multivariable gamma model was constructed to study the association between length of average time of DSMES and year after holding all covariates constant. A gamma regression model was chosen because the distribution of length of DSMES time was non-negative and positively skewed. Data management and analysis were conducted using SAS statistical software version 9.4 (SAS Institute Inc.). A $p$-value less than 0.05 was considered statistically significant.

## Results

Table 1 provides the distributions of cancer survivors' characteristics using 2006, 2012, and 2019 as examples. In 2006, 8,312 cancer survivors were identified as being eligible for this study, 30,134 in 2012 and 39,363 in 2009. Information on Medicaid-Medicare dual eligibility was not available until 2007 and around 3.84% of individuals who have both Medicare and Medicaid in 2012 and 6.10% in 2019.

The portions of cancer survivors who are female, older than 85, diagnosed with breast cancer, diagnosed with Type 2 diabetes only, not disabled, living in areas with very good education, living under very low poverty level, living in a metropolitan area, and having a higher comorbidity index have been increased over time. However, the percentage of diabetic patients with complications or having dual eligibility has been decreased. These trends may provide policymakers and healthcare providers with a better understanding of the trend of prostate, breast, and colorectal cancer survivors' characteristics.

Table 2 demonstrates the number of people who first-time used the DSMES, the total number of all people who have used the DSMES in each particular year, the total number of DSMES sessions used by cancer survivors, and the average number of minutes of DSMES used by cancer survivors each year. Over 14 years, the number of first-time users has slowly increased by almost 8 folds (from 99 to 769) for the first ten years but suddenly dropped starting 2016. On the contrary, the total number of users, the number of DSMES sessions attended by these users, and the average time of DSMES participation have increased and maintained on a plateau after 2016.

Fig 1 shows the ratio of first-time users per 1000 and the ratio of all users (first-time and follow-up users) per 1000. Similar to the finding in Table 1, while the number of people who newly started DSMES has decreased since 2016, more previous users stayed in the program.

Table 3 illustrates the odds of being a first-time DSMES user by a variety of cancer survivors' characteristics. Using the year of 2006 as the reference, the odds of having first-time DSMES among diabetic cancer survivors scaled up and down until 2015 (O.R. = 2.26, 95% CI: 1.81–2.81) where the peak was reached, and the trend gradually decreased after 2015. The table also shows that older people were less likely to initiate the first DSMES (O.R. = 0.36, $p<0.0001$). Other barriers include living in metropolitan areas and having a higher comorbidity index. Next, Hispanic populations were more likely to initiate the use than white populations (O.R. = 1.19, $p<0.001$). Other facilitating factors include people who were just diagnosed with diabetes, had complications or had dual eligibility. The multivariable model was stratified by three types of cancers (S1-S3 Tables in S1 File). The major findings remain the same except for that among colorectal cancer survivors, no significant influence was found from non-metro residence and comorbidity index on the first-time DSMES utilization.

**Table 1. Characteristics of study sample in 2006, 2012, and 2019.**

| Year Characteristics, N (%) | 2006 (N = 8312) | 2012 (N = 30134) | 2019 (N = 39363) |
|---|---|---|---|
| **Gender** | | | |
| Male | 4,600 (55.34) | 16,433 (54.53) | 20,165 (51.23) |
| Female | 3,712 (44.66) | 13,701 (45.47) | 19,198 (48.77) |
| **Age Group** | | | |
| 66–74 | 3,342 (40.21) | 10,299 (34.18) | 13,015 (33.06) |
| 75–84 | 3,909 (47.03) | 14,258 (47.32) | 17,930 (45.55) |
| 85+ | 1,061 (12.76) | 5,577 (18.51) | 8,418 (21.39) |
| **Race/Ethnicity** | | | |
| Non-Hispanic White | 5,788 (69.63) | 20,995 (69.67) | 27,523 (69.92) |
| Non-Hispanic Black | 852 (10.25) | 3,184 (10.57) | 4,029 (10.24) |
| Hispanic | 1,577 (18.97) | 5,406 (17.94) | 6,795 (17.26) |
| Others | 95 (1.14) | 549 (1.82) | 1,016 (2.58) |
| **Cancer Type** | | | |
| Prostate cancer | 3,731 (44.89) | 13,648 (45.29) | 16,720 (42.48) |
| Breast cancer | 2,923 (35.17) | 11,033 (36.61) | 16,324 (41.47) |
| Colorectal cancer | 1,658 (19.95) | 5,453 (18.10) | 6,319 (16.05) |
| **Diabetes Type** | | | |
| Type I only | 451 (5.43) | 1,081 (3.59) | 899 (2.28) |
| Type II only | 7,856 (94.51) | 29,028 (96.33) | 38,433 (97.64) |
| Both | 5 (0.06) | 25 (0.08) | 31 (0.08) |
| **Newly Diagnosed Diabetes** | 632 (7.60) | 1,266 (4.20) | 806 (2.05) |
| **Diabetes with Complications** | 5,211 (62.69) | 17,388 (57.70) | 18,575 (47.19) |
| **Medicare Original Entitlement** | | | |
| Disabled | 708 (8.52) | 2,527 (8.39) | 2,469 (6.27) |
| Non-Disabled | 7,604 (91.48) | 27,607 (91.61) | 36,894 (93.73) |
| **Medicare-Medicaid Dual Eligibility** | 1,449 (17.43) | 4,787 (15.89) | 4,416 (11.22) |
| **Quartiles of Percent Living in Regions with < 12 Years of Education** | | | |
| [0–11] (Very good education) | 1,797 (22.59) | 7,320 (25.74) | 11,685 (31.59) |
| [12–20] (Good education) | 1,851 (23.27) | 6,844 (24.06) | 9,331 (25.23) |
| [21–29] (Poor education) | 2,035 (25.58) | 7,025 (24.70) | 8,347 (22.56) |
| [30–100] (Very poor education) | 2,273 (28.57) | 7,254 (25.50) | 7,628 (20.62) |
| **Quartiles of Percent Living Below Poverty Level** | | | |
| [0–7] (Very low poverty) | 1,625 (20.42) | 6,921 (24.33) | 11,283 (30.50) |
| [8–11] (Low poverty) | 1,879 (23.62) | 6,890 (24.22) | 9,240 (24.98) |
| [12–18] (High poverty) | 2,205 (27.71) | 7,619 (26.79) | 9,071 (24.52) |
| [19–100] (Very high poverty) | 2,247 (28.24) | 7,013 (24.66) | 7,397 (20.00) |
| **Residence** | | | |
| Metro | 6,571 (79.05) | 24,484 (81.25) | 32,607 (82.84) |
| Non-Metro | 1,741 (20.95) | 5,650 (18.75) | 6,756 (17.16) |
| **Elixhauser Comorbidity Index*** | | | |
| [0–1] | 2,314 (27.84) | 6,644 (22.05) | 6,553 (16.65) |
| [2–3] | 3,043 (36.61) | 10,921 (36.24) | 13,211 (33.56) |
| [>3] | 2,955 (35.55) | 12,569 (41.71) | 19,599 (49.79) |

*Excludes diabetes, lymphoma, metastatic cancer, solid tumor without metastasis

**Table 2. Trend of numbers of users and sessions from 2006 to 2019.**

| Year (# Cancer Survivors) | # of First-Time Users | # of All Users | Total # of DSMT Sessions | DSMT Time (mean ± std. and range) |
|---|---|---|---|---|
| 2006 (n = 8312) | 99 | 123 | 181 | 44 ± 24 (28–60) |
| 2007 (n = 11836) | 123 | 169 | 256 | 45 ± 24 (29–62) |
| 2008 (n = 15632) | 163 | 231 | 355 | 46 ± 24 (30–62) |
| 2009 (n = 19534) | 155 | 229 | 391 | 51 ± 29 (32–71) |
| 2010 (n = 23393) | 165 | 271 | 459 | 51 ± 30 (31–71) |
| 2011 (n = 27187) | 168 | 286 | 500 | 52 ± 33 (30–75) |
| 2012 (n = 30134) | 193 | 324 | 562 | 52 ± 31 (31–73) |
| 2013 (n = 31731) | 168 | 318 | 569 | 54 ± 33 (31–76) |
| 2014 (n = 34346) | 290 | 442 | 736 | 50 ± 31 (29–71) |
| 2015 (n = 35871) | 769 | 1,009 | 2,528 | 75 ± 51 (41–109) |
| 2016 (n = 38007) | 389 | 1,181 | 3,380 | 86 ± 60 (46–126) |
| 2017 (n = 39234) | 358 | 1,201 | 3,540 | 88 ± 62 (47–130) |
| 2018 (n = 38899) | 308 | 1,165 | 3,518 | 91 ± 63 (48–133) |
| 2019 (n = 39363) | 294 | 1,175 | 3,477 | 89 ± 61 (48–130) |

Table 4 is focused on which characteristics are associated with a longer DSMES utilization (in minutes) among all users. Like the findings in Table 2 and Fig 1, there is an increasing use of the DSMES in minutes especially after 2015 ($p<0.001$). Next, gender, age, cancer type, and presence of diabetes complications are not significantly associated with the length of DSMES utilization. Third, people who were Hispanic (O.R. = 1.10), disabled (O.R. = 1.08), have both Medicare and Medicaid (O.R. = 1.25), living in areas with very poor education (O.R. = 1.09–1.21), and having more comorbidities (O.R. = 1.06–1.14) were more likely to use the DSMES for a longer term. However, people living in metropolitan areas were less likely to have a longer DSMES utilization (O.R. = 0.90, $p <0.0001$).

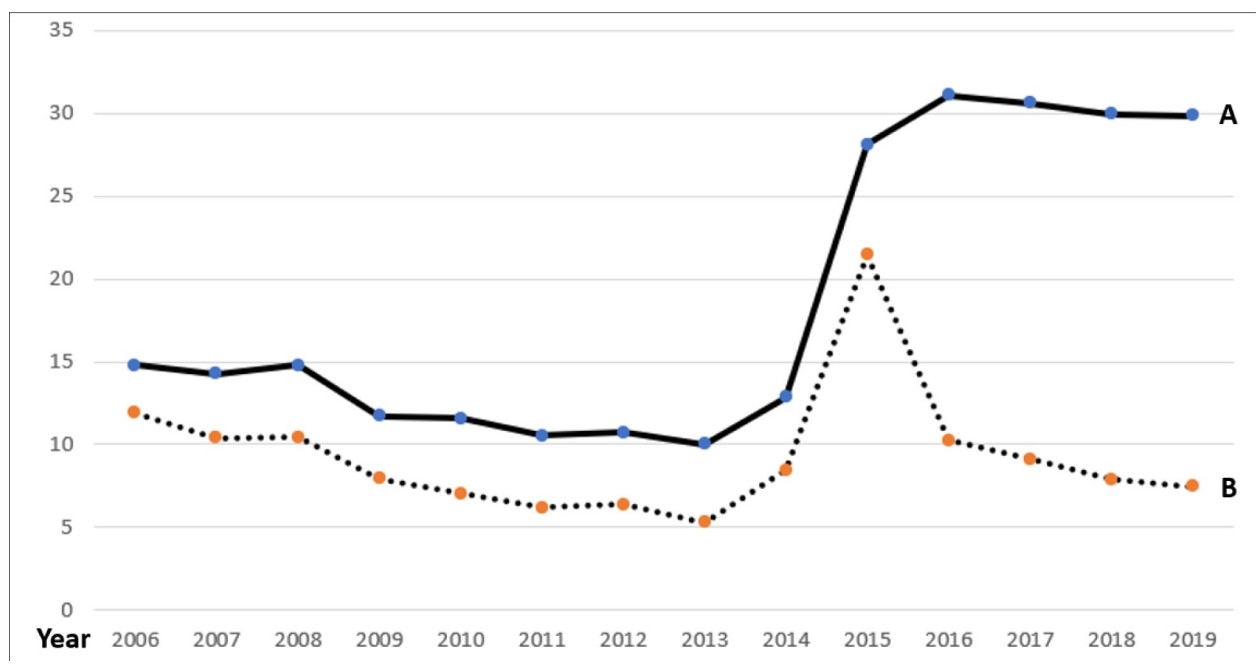

**Fig 1. Trends of first-time DSMT users and the total number of users (per 1000).** (A) Accumulated Number of DSMT Users per 1000. (B) Number of First-Ever DSMT Users per 1000.

**Table 3. Model predicting the first-time DSMT use.**

| Variables | First DSMT Use Rate | OR* | 95% CI | | p-value |
|---|---|---|---|---|---|
| **Year** | | | | | |
| 2006 (Referent) | 181/8,312 | 1.00 | - | - | - |
| 2007 | 256/11,836 | 0.91 | 0.69 | 1.20 | 0.5024 |
| 2008 | 355/15,632 | 0.93 | 0.72 | 1.20 | 0.5753 |
| 2009 | 391/19,534 | 0.75 | 0.58 | 0.98 | 0.0338 |
| 2010 | 459/23,393 | 0.67 | 0.52 | 0.87 | 0.0027 |
| 2011 | 500/27,187 | 0.60 | 0.47 | 0.78 | 0.0001 |
| 2012 | 562/30,134 | 0.65 | 0.50 | 0.83 | 0.0007 |
| 2013 | 569/31,731 | 0.53 | 0.41 | 0.69 | < .0001 |
| 2014 | 736/34,346 | 0.87 | 0.69 | 1.11 | 0.2592 |
| 2015 | 2,528/35,871 | 2.26 | 1.81 | 2.81 | < .0001 |
| 2016 | 3,380/38,007 | 1.09 | 0.86 | 1.37 | 0.4862 |
| 2017 | 3,540/39,234 | 1.02 | 0.81 | 1.29 | 0.8672 |
| 2018 | 3,518/38,899 | 0.88 | 0.69 | 1.11 | 0.284 |
| 2019 | 3,477/39,363 | 0.88 | 0.69 | 1.12 | 0.2929 |
| **Gender** | | | | | |
| Male (Referent) | 1,852/210,727 | 1.00 | - | - | - |
| Female | 1,790/182,752 | 1.14 | 0.98 | 1.34 | 0.0992 |
| **Age Group** | | | | | |
| 66–74 (Referent) | 1,841/136,237 | 1.00 | - | - | - |
| 75–84 | 1,440/182,081 | 0.61 | 0.56 | 0.65 | < .0001 |
| 85+ | 361/75,161 | 0.36 | 0.32 | 0.41 | < .0001 |
| **Race/Ethnicity** | | | | | |
| Non-Hispanic White (Referent) | 2,366/276,030 | 1.00 | - | - | - |
| Non-Hispanic Black | 403/40,273 | 1.03 | 0.92 | 1.16 | 0.5715 |
| Hispanic | 801/69,357 | 1.19 | 1.09 | 1.31 | 0.0002 |
| Other | 72/7,819 | 1.03 | 0.81 | 1.31 | 0.7874 |
| **Cancer Type** | | | | | |
| Prostate (Referent) | 1,539/174,561 | 1.00 | - | - | - |
| Breast | 1,492/150,186 | 0.97 | 0.82 | 1.15 | 0.7137 |
| Colorectal | 611/68,732 | 0.98 | 0.87 | 1.11 | 0.7755 |
| **Newly Diagnosed Diabetes** | 515/15,741 | 4.70 | 4.23 | 5.21 | < .0001 |
| **Diabetes with Complications (Referent: No complications)** | 2,324/216,055 | 1.71 | 1.59 | 1.84 | < .0001 |
| **Medicare Original Entitlement: Disabled (Referent: Not disabled)** | 294/29,958 | 0.90 | 0.79 | 1.02 | 0.0939 |
| **Dual Eligibility (Referent: No dual eligibility)** | 640/55,016 | 1.26 | 1.14 | 1.40 | < .0001 |
| **Quartiles of Percent Living in Regions with < 12 Years of Education** | | | | | |
| [0–11] (Referent) (Very good education) | 918/102,727 | 1.00 | - | - | - |
| [12–20] (Good education) | 787/91,091 | 0.96 | 0.87 | 1.06 | 0.4183 |
| [21–29] (Poor education) | 812/88,822 | 0.98 | 0.88 | 1.08 | 0.6462 |
| [30–100] (Very poor education) | 891/88,719 | 1.02 | 0.93 | 1.13 | 0.6407 |
| **Metropolitan Areas (Referent: Non-Metro)** | 2,876/320,879 | 0.83 | 0.76 | 0.90 | < .0001 |
| **Elixhauser comorbidity index** | | | | | |
| [0–1] (Referent) | 847/81,988 | 1.00 | - | - | - |
| [2–3] | 1,381/139,565 | 1.01 | 1.11 | 2.25 | 0.8108 |
| [>3] | 1,414/171,926 | 0.87 | 0.95 | 1.00 | 0.0027 |

*OR = Odds Ratio.

**Table 4. Model predicting the average time of DSMT use (in minutes) among all users.**

| Variables | Estimate | 95% CI | | p-value |
|---|---|---|---|---|
| **Year** | | | | |
| 2006 (Referent) | 1.00 | - | - | - |
| 2007 | 1.03 | 0.90 | 1.18 | 0.6312 |
| 2008 | 1.05 | 0.93 | 1.19 | 0.434 |
| 2009 | 1.16 | 1.02 | 1.31 | 0.0217 |
| 2010 | 1.14 | 1.00 | 1.29 | 0.0421 |
| 2011 | 1.16 | 1.02 | 1.31 | 0.0209 |
| 2012 | 1.14 | 1.01 | 1.29 | 0.0319 |
| 2013 | 1.18 | 1.04 | 1.33 | 0.0088 |
| 2014 | 1.08 | 0.96 | 1.22 | 0.1888 |
| 2015 | 1.41 | 1.27 | 1.58 | < .0001 |
| 2016 | 1.59 | 1.43 | 1.77 | < .0001 |
| 2017 | 1.60 | 1.43 | 1.78 | < .0001 |
| 2018 | 1.65 | 1.48 | 1.84 | < .0001 |
| 2019 | 1.62 | 1.46 | 1.81 | < .0001 |
| **Sex** | | | | |
| Male (Referent) | 1.00 | - | - | - |
| Female | 1.02 | 0.93 | 1.11 | 0.6628 |
| **Age Group** | | | | |
| 66–74 (Referent) | 1.00 | - | - | - |
| 75–84 | 0.98 | 0.94 | 1.01 | 0.2017 |
| 85+ | 1.06 | 0.99 | 1.14 | 0.0993 |
| **Race/Ethnicity** | | | | |
| Non-Hispanic White (Referent) | 1.00 | - | - | - |
| Non-Hispanic Black | 0.96 | 0.90 | 1.03 | 0.2417 |
| Hispanic | 1.10 | 1.05 | 1.16 | 0.0002 |
| Other | 1.06 | 0.91 | 1.24 | 0.4455 |
| **Cancer type** | | | | |
| Prostate (Referent) | 1.00 | - | - | - |
| Breast | 1.03 | 0.94 | 1.13 | 0.5545 |
| Colorectal | 1.02 | 0.95 | 1.09 | 0.5759 |
| **Newly Diagnosed Diabetes** | 0.97 | 0.91 | 1.02 | 0.236 |
| **Diabetes with Complications (Referent: No complications)** | 1.03 | 0.99 | 1.07 | 0.1445 |
| **Medicare Original Entitlement: Disabled (Referent: Not disabled)** | 1.08 | 1.01 | 1.16 | 0.0266 |
| **Dual Eligibility (Referent: No dual eligibility)** | 1.25 | 1.18 | 1.31 | < .0001 |
| **Quartiles of Percent Living in Regions with < 12 Years of Education** | | | | |
| [0–11] (Referent) (Very good education) | 1.00 | - | - | - |
| [12–20] (Good education) | 1.09 | 1.02 | 1.15 | 0.0065 |
| [21–29] (Poor education) | 1.21 | 1.14 | 1.28 | < .0001 |
| [30–100] (Very poor education) | 1.20 | 1.13 | 1.27 | < .0001 |
| **Metropolitan Areas (Referent: Non-Metro)** | 0.90 | 0.85 | 0.94 | < .0001 |
| **Elixhauser comorbidity index** | | | | |
| [0–1] (Referent) | 1.00 | - | - | - |
| [2–3] | 1.06 | 1.02 | 1.11 | 0.0043 |
| [>3] | 1.14 | 1.08 | 1.19 | < .0001 |

## Discussion

This study is one of the first to assess the trend of the first-time and follow-up utilization of the DSMES among three types of cancer survivors in Texas from 2006 to 2019. Five important findings are discovered: (1) The number of cancer survivors per year has consistently increased from 2006 to 2019. (2) The number of first-time users and the number of DSMES sessions attended increased for the first ten years but decreased after 2016. (3) The length of DSMES participation among users increased during 2013 and 2016 and remained on a plateau since 2016. (4) The facilitators of the first-time DSMES utilization include Hispanic identity, newly diagnosed with diabetes, having diabetes complications, and having dual eligibility. The barriers to DSMES initiation include being older than 75, living in metropolitan areas, and having more than three comorbidities. (5) The facilitators of the overall DSMES utilization are Hispanic identity, having disability, having dual eligibility, living in an area with poor education, and having more than three comorbidities. The barrier to DSMES retention is living in a metropolitan area with more than 250,000 residents.

### Characteristics of cancer survivors

The study identified Medicare beneficiaries that met the inclusion criteria every year and accumulated eligible samples from 2006 to 2019. Given the nature of aging, the later sample contains older people with more comorbidities. Additionally, this study shows that the portions of cancer survivors who live in areas with very good education, very low poverty, and metropolitan areas have increased over 14 years. These findings are in agreement with a rich body of literature regarding socially induced disparities in cancer survival. Studies show that living in rural areas or having poor access to education, employment, healthcare, insurance, quality housing, food security, and social support adversely contribute to cancer survival rates [25, 26]. Cancer patients have complex healthcare needs, and it is critical to advance equity in cancer health through a better understanding of, and action to address, social determinants [27]. Such a blueprint offers insights on practice, research, and policy to reduce the burden of cancers by addressing income inequalities, supporting targeted policies with explicit benefits for disadvantaged groups, and incentivizing healthcare providers to adopt holistic healthcare approach. More longitudinal studies are also recommended to track the improvement in cancer survivorship of disadvantaged populations.

### Trends of initial DSMES and DSMES follow-up

Since each cancer survivor in the study could join DSMES at any given time between 1999 and 2019, the number of first-time users provides insight on how available and accessible the DSMES program is for all cancer survivors. On the other hand, the accumulated number of all users each year indicates the retention of all qualified cancer survivors. This study discovered a rapid increase in the number of first-time DSMES users in 2015. This might be explained by the increase in the number of certified diabetes educators and CMS's leading the trend of healthcare systems to shift from fee-for-service to value-based care over the past decade [28, 29]. However, more studies are needed to investigate the cause of the increase in 2015 in Texas particularly. The study findings show a sudden drop in the trend of initiation in 2016, whereas the retention has remained on a plateau following a substantial increase in 2013. This might be explained by the change in the Medicare reimbursement for Chronic Care Management (CCM) in 2016 to incentivize a spending on a longer time on CCM services [30]. As a result, DSMES providers (e.g., diabetes education specialists) might focus more on current users to maximize the amount of reimbursement instead of attracting new users. By learning that the DSMT utilization is very low among Medicare beneficiaries, strategies to encourage diabetes

care and education specialists to improve the enrollment and retention among cancer survivors are highly important to ensure the successful implementation of the program. More research is needed to explain changes in both DSMES initiation and retention trends by Medicare beneficiary cancer survivors in order to inform reimbursement strategies.

## Determinants of utilization

This study found that DSMES initiation is positively associated with Hispanic ethnicity, diabetes incidence, diabetes complications, and dual eligibility, but negatively associated with older age, living in metropolitan areas, and having more comorbidities. The finding about residence is inconsistent with the literature, noting that urban residents (areas >50,000 residents) have more DSMES participation [31]. However, this study defined "metropolitan" as an area with more than 250,000 residents, and the study was focused on cancer survivors, not Medicare beneficiaries in general [32]. Next, a 2012 report suggested that fewer African American and Hispanic beneficiaries know that Medicare helps pay for diabetes self-management education, yet this study pointed out that Hispanic ethnicity is a favorable factor for both initiation (first-time use) and retention (average minutes per use) [33]. A recent finding revealed that Latino participants are more likely to complete DSMES sessions because they are enrolled into culturally tailored programs [34]. Therefore, developing programs unique to each culture is highly recommended in enhancing DSMES participation among racial minorities with diabetes. Finally, the study results show that dually eligible individuals are more likely to be first-time and follow-up users. It might be explained by their better awareness and access to more providers and DSMES programs covered by both Medicare and Medicaid. This finding may guide future efforts to strengthen DSMES utilization by increasing availability and payment of evidence-based practices.

Different from the determinants of initiation, this study found that having disability, living in areas with poor education, and having multiple comorbidities are related to a longer time of DSMES use. One study pointed out that people with higher education are more likely to work full or part-time, that impeded their continuation in the program [35]. On the other hand, individuals with disability or complications might be more likely to have routine medical visits, that increases the likelihood to stay in health systems and use DSMT [36]. Because the former research drew conclusions from various samples, more research on factors to retain DSMES participants would be helpful to design strategies to support them to complete the program.

DSMT/DSMES has been adopted by Medicare since 1997 and it is an evidence-based program that educates people living with diabetes on how to mitigate the burden of this chronic condition and reduce the risk of complication [37]. However, its utilization among people living with diabetes generally and among those cancer survivors living with diabetes is considerably limited. On average, this study found that around 10 of 1000 cancer survivors per year started to participate in DSMES over 14 years. A 2015 study also shows that only 5% of Medicare beneficiaries attended DSMES [7]. The Centers for Disease Control and Prevention identified three barriers to DSMT or DSMES utilization including programmatic barriers to starting or sustaining services, provider barriers to referral, and barriers to access and participation [38]. In 2022, the reimbursement to a provider in Galveston, TX for a 30-minute DSMES individual session is only $56.88, and $16.22 per patient for a 30-minute group session [39]. While DSMES will greatly increase the value of diabetic care, it has not been reasonably reimbursed in any value-based care model [40]. Additionally, prostate, breast, and colorectal cancer survivors account for 60% of all survivors [41]. The evidence-based DSMES that combines coping skills and lifestyle changes is very likely to enhance cancer survivorship [15–18]. In response, this study highly supports better coordination between primary care practitioners,

oncologists, and endocrinologists to follow up with cancer survivors in a course covering diabetes [42, 43]. Strategies to address the three aforementioned barriers are also critical such as (1) individual level: use of telehealth to overcome transportation barriers, (2) provider level: incentives for providers to refer eligible patients to DSMES programs, and (3) system level: a reimbursement scheme to incentivize healthcare institutions to include DSMES in survivorship care plans [40, 43].

## Limitations

This study has several limitations. First, the study used administrative claims data of Texas Medicare beneficiaries who had survived breast, prostate, and colorectal cancers for at least five years. The results cannot be generalized to other types of cancer survivors or Medicare beneficiaries in other states. Second, to qualify for DSMES, an individual must not only have a diagnosis of Type 1 or Type 2 diabetes but also have a fasting blood glucose of 126 mg/dL on two separate occasions [44]. However, the Medicare claims data lack the information on the second criterion. As this study has assumed that all cancer survivors diagnosed with diabetes qualify for the program, the participation rate calculated in this study could seem too low. Third, the American Academy of Professional Coders argues that diabetes is frequently under-coded which might result in lack of referral to DSMES [45]. Accurate coding of diabetes and its complications will render necessary services to cancer survivors.

## Conclusion

The study is one of the first to assess the trend of the first-time and follow-up utilization of the DSMES among three types of cancer survivors in Texas from 2006 to 2019. A comprehensive plan of care for cancer survivors with diabetes is critical in stimulating utilization and ensuring full payment. The decrease in the initiation and the increase in the continuation may guide future efforts to provide more incentives to healthcare providers to consider DSMES as an essential component of cancer care. Additionally, the study discovered that Hispanic ethnicity is positively related to initiation and retention. Availability and accessibility of culturally tailored programs might explain this finding, but more studies are needed to inform strategies in order to target other racial minorities.

## Supporting information

**S1 File. Contains all the supporting tables (S1–S3 Tables) and figure (S1 Fig).**
(DOCX)

## Author Contributions

**Conceptualization:** Wei-Chen Lee, Hani Serag, Hanaa Sallam, Yong-Fang Kuo.

**Data curation:** Biai Dominique Elmir Digbeu.

**Formal analysis:** Biai Dominique Elmir Digbeu.

**Funding acquisition:** Wei-Chen Lee, Yong-Fang Kuo.

**Investigation:** Wei-Chen Lee.

**Methodology:** Wei-Chen Lee, Yong-Fang Kuo.

**Software:** Yong-Fang Kuo.

**Validation:** Wei-Chen Lee.

**Visualization:** Biai Dominique Elmir Digbeu.

**Writing – original draft:** Wei-Chen Lee, Hani Serag, Hanaa Sallam.

**Writing – review & editing:** Wei-Chen Lee, Biai Dominique Elmir Digbeu, Hani Serag, Hanaa Sallam, Yong-Fang Kuo.

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
