## [Decision Letter · Decision Letter 0]

20 Jun 2023

PONE-D-23-10647Utilization of Diabetes Self-Management Program among Breast, Prostate, and Colorectal Cancer Survivors: Using 2006-2019 Texas Medicare DataPLOS ONE

Dear Dr. UTMB,

Thank you for submitting your manuscript to PLOS ONE. After careful consideration, we feel that it has merit but does not fully meet PLOS ONE’s publication criteria as it currently stands. Therefore, we invite you to submit a revised version of the manuscript that addresses the points raised during the review process.

We look forward to receiving your revised manuscript.

Kind regards,

Meng Li

Academic Editor

PLOS ONE

“The study was made possible by the grant (#RP210130) from Cancer Prevention and Research Institute of Texas (CPRIT) Data Management and Analysis Core (DMAC). The funder had no role in study design, data collection and analysis, decision to public, or preparation of the manuscript.”

“The study was made possible by the grant (#RP210130) from Cancer Prevention and Research Institute of Texas (CPRIT) Data Management and Analysis Core (DMAC). The funder had no role in study design, data collection and analysis, decision to public, or preparation of the manuscript.”

Reviewers' comments:

Reviewer's Responses to Questions

**Comments to the Author**

1. Is the manuscript technically sound, and do the data support the conclusions?

Reviewer #1: Yes

2. Has the statistical analysis been performed appropriately and rigorously? 

Reviewer #1: Yes

3. Have the authors made all data underlying the findings in their manuscript fully available?

Reviewer #1: Yes

4. Is the manuscript presented in an intelligible fashion and written in standard English?

Reviewer #1: Yes

5. Review Comments to the Author

Reviewer #1: Thanks for the opportunity to review this manuscript. The present analysis is a retrospective study of DSMT utilization among breast, prostate, and colorectal cancer survivors and associated patients’ characteristics in Texas using the registry-claims linkage data. The findings and the message of this study are valuable and informative to the journal audience. However, there are some concerns regarding the measurement of variables and delivery of findings. These concerns and other comments are listed below.

Introduction

• Diabetes is a board topic. As the study is focusing on cancer survivors, the authors should include evidence related to diabetes more specific to cancer patients other than the general population to elaborate why this study is significant. For example, what is the prevalence/incidence of diabetes among the cancer population? Will the survivorship be affected by diabetes and diabetes complications? Does diabetes management improve the quality of life for cancer survivors?...

• I would suggest describing more details regarding the services and provider information of the DSMT program.

Methods

• “In 2021, the TCR joined the SEER program.” is unrelated information which can be deleted.

• Were patients who had severe renal diseases or kidney transplantation excluded?

• How was dual eligibility imputed before 2007 in the regression analysis?

• What was the data source for the zip code level data (education & income)?

• Poverty level was included in the descriptive analysis but not introduced in the “Independent Variable & Covariates” section.

• How was metro/non-metro classified?

Results

• Please use 1000 separator in the text and tables to be more reader friendly.

Discussion

• The statement “the number of cancer survivors has increased almost five-fold and the number of first-time DSMES users increased almost eight-fold” is problematic. The denominators are different. For instance, the 2006 cohort is for cancer patients diagnosed in 1999-2001 and the 2019 cohort is for 1999-2014. The numbers are not comparable.

• Please consider discussing the determinants of average time of DSMT use.

6. PLOS authors have the option to publish the peer review history of their article (what does this mean?). If published, this will include your full peer review and any attached files.

Reviewer #1: No

---

## [Author Response · Author response to Decision Letter 0]

29 Jun 2023

We are delighted to learn that the reviewer found our study valuable and informative to the journal audience. We have addressed the concerns about the literature and definitions of variables. We also expanded our discussion section, and all changes could be found in the revised manuscript. Please do not hesitate to let us know if further revision is needed to enhance the quality of our manuscript. Thank you.

---

## [Editor Report · Decision Letter 1]

17 Jul 2023

Utilization of Diabetes Self-Management Program among Breast, Prostate, and Colorectal Cancer Survivors: Using 2006-2019 Texas Medicare Data

PONE-D-23-10647R1

Dear Dr. Lee,

We’re pleased to inform you that your manuscript has been judged scientifically suitable for publication and will be formally accepted for publication once it meets all outstanding technical requirements.

Kind regards,

Meng Li

Academic Editor

PLOS ONE
---

## [Editor Report · Acceptance letter]

19 Jul 2023

PONE-D-23-10647R1 

Utilization of Diabetes Self-Management Program among Breast, Prostate, and Colorectal Cancer Survivors: Using 2006-2019 Texas Medicare Data 

Dear Dr. Lee:

I'm pleased to inform you that your manuscript has been deemed suitable for publication in PLOS ONE. Congratulations! Your manuscript is now with our production department. 

Kind regards, 

on behalf of

Dr. Meng Li 

Academic Editor

PLOS ONE